# L-Pampo™, a Novel TLR2/3 Agonist, Acts as a Potent Cancer Vaccine Adjuvant by Activating Draining Lymph Node Dendritic Cells

**DOI:** 10.3390/cancers15153978

**Published:** 2023-08-04

**Authors:** Yoonki Heo, Eunbyeol Ko, Sejung Park, Si-On Park, Byung-Cheol Ahn, Jung-Sun Yum, Eunyoung Chun

**Affiliations:** R&D Center, CHA Vaccine Institute, Seongnam-si 13493, Republic of Korea; hyk0345@chamc.co.kr (Y.H.); Eunbyeol.ko@ivi.int (E.K.); parksj@chamc.co.kr (S.P.); sionirena@chamc.co.kr (S.-O.P.); bcahn@chamc.co.kr (B.-C.A.); jsyum@chamc.co.kr (J.-S.Y.)

**Keywords:** cancer vaccine, TLR agonist, dendritic cell (DC), draining lymph node, anti-tumor efficacy, immune checkpoint inhibitor, combination immunotherapy

## Abstract

**Simple Summary:**

Toll-like receptor (TLR) agonists induce strong immune responses, which are used as an effective adjuvant system for vaccine development. L-pampo™ is our innovative adjuvant system containing TLR2 and TLR3 agonists, and it provides robust humoral and cellular immune responses against infectious diseases. Cancer vaccines are a type of cancer immunotherapy that activate the host immune system to recognize and eliminate cancer cells. Here, we investigate the anti-tumor efficacy and the mechanism of action of vaccines formulated with L-pampo™. We demonstrate that L-pampo™ directly stimulates the maturation and function of dendritic cells, which are essential antigen-presenting cells. Additionally, the vaccine formulated with L-pampo™ induces the recruitment of dendritic cells into lymph nodes, where they prime antigen-specific T-cell responses and induce potent anti-tumor effects. Interestingly, combining vaccines formulated with L-pampo™ with immune checkpoint inhibitors induces a synergistic anti-tumor effect, indicating that L-pampo™ can be a powerful cancer vaccine adjuvant and a potent partner for combination immunotherapy.

**Abstract:**

TLR agonists have emerged as an efficient cancer vaccine adjuvant system that induces robust immune responses. L-pampo™, a proprietary vaccine adjuvant of TLR2 and TLR3 agonists, promotes strong humoral and cellular immune responses against infectious diseases. In this study, we demonstrate that vaccines formulated with L-pampo™ affect the recruitment and activation of dendritic cells (DCs) in draining lymph nodes (dLNs) and leading to antigen-specific T-cell responses and anti-tumor efficacy. We analyzed DC maturation and T-cell proliferation using flow cytometry and ELISA. We determined the effect of L-pampo™ on DCs in dLNs and antigen-specific T-cell responses using flow cytometric analysis and the ELISPOT assay. We employed murine tumor models and analyzed the anti-tumor effect of L-pampo™. We found that L-pampo™ directly enhanced the maturation and cytokine production of DCs and, consequently, T-cell proliferation. OVA or OVA peptide formulated with L-pampo™ promoted DC migration into dLNs and increased activation markers and specific DC subsets within dLNs. In addition, vaccines admixed with L-pampo™ promoted antigen-specific T-cell responses and anti-tumor efficacy. Moreover, the combination of L-pampo™ with an immune checkpoint inhibitor synergistically improved the anti-tumor effect. This study suggests that L-pampo™ can be a potent cancer vaccine adjuvant and a suitable candidate for combination immunotherapy.

## 1. Introduction

Cancer immunotherapy, which includes cancer vaccines, immune checkpoint inhibitors, and adoptive cell therapy, has emerged as a promising approach to treating cancer [1]. Cancer vaccines aim to stimulate the host immune system to selectively recognize and eliminate cancer cells [2]. Multiple cancer vaccine platforms have been developed, including cell-based, peptide- or protein-based, viral-based, and nucleic acid-based vaccines to elicit CD8^+^ cytotoxic T lymphocyte (CTL), a pivotal effector cell in tumor immunity. Peptide or protein cancer vaccines have the advantages of safety, less toxicity, ease of production, and targeting multi-antigens and epitopes [3]. However, peptide- or protein-based vaccines often show less immunogenicity and poor antigen presentation due to the absence of costimulatory molecules and danger signals derived from matured antigen-presenting cells (APCs). To improve overall immunogenicity and activate specific CD8^+^ T-cell responses through APC activation, a powerful adjuvant system that can attract immune cells to the site of injection, promote cell-mediated trafficking antigens to drain lymph nodes, and trigger APC activation is indispensable for peptide or protein cancer vaccine development [4].

Montanide ISA 51 and granulocyte-macrophage colony-stimulating factor (GM-CSF) have been widely used as adjuvants for cancer peptide vaccines. Montanide ISA 51 is a water-in-oil emulsion that forms a depot of antigens at the injection site, leading to protection and a slow release of antigens [5]. GM-CSF activates innate immune cells, such as dendritic cells and macrophages, that initiate adaptive immune responses [6]. Alum (aluminum salts) is an approved human vaccine adjuvant that promotes antigen uptake and presentation. Alum also creates a depot effect that can prolong the exposure of antigens to the immune system and increase the duration of the immune response. Recent studies have reported that alum can also induce anti-tumor efficacy when used on the nanoscale [7,8]. Toll-like receptor (TLR) agonists have recently emerged as notable adjuvants for cancer vaccines since they can increase vaccine efficacy by mimicking microbial stimulation. Several TLR agonists have been tested in clinical trials for various types of cancers, including lymphoma, melanoma, non-small cell lung cancer (NSCLC), prostate cancer, and glioblastoma [9].

We have developed a proprietary adjuvant, L-pampo™, composed of TLR1/2 and TLR3 agonists. L-pampo™ adjuvanted subunit protein vaccines induce robust humoral and cellular immune responses against HBV and SARS-CoV-2 [10,11]. Most recently, we also identified the therapeutic role of L-pampo™ per se, which promotes anti-tumor immunity and enhances immune checkpoint blockade [12]. However, whether L-pampo™ can be a potent cancer vaccine adjuvant inducing anti-tumor efficacy has not been investigated.

This study showed that L-pampo™ directly induced DC maturation and function. L-pampo™ promoted DC recruitment into draining lymph nodes and expansion of DC subsets compared to Montanide ISA 51, GM-CSF, or Alum. The OVA or OVA peptide admixed with L-pampo™ induced antigen-specific T-cell immune responses in an in vivo system. In mouse tumor models, OVA or OVA peptide admixed with L-pampo™ significantly inhibited tumor growth and increased the survival rate. Furthermore, the combination of L-pampo™ adjuvanted OVA vaccine with an immune checkpoint inhibitor manifested synergistic anti-tumor efficacy. Our study demonstrates that L-pampo™ is a promising adjuvant for cancer vaccines and an effective combination partner to ICIs for cancer immunotherapy.

## 2. Materials and Methods

### 2.1. Animals

Female C57BL/6 (6–8 weeks old) mice were purchased from Orient Bio Inc. (Seongnam, Korea) and housed in a specific-pathogen-free animal facility at CHA University (Seongnam, Korea). All mice were handled in accordance with the standards approved by the Institutional Animal Care and Use Committee (IACUC) of CHA University (IACUC; #220089).

### 2.2. Cancer Cell Lines

A B16F10-OVA murine melanoma cell was kindly provided by Prof. Seung-Woo Lee (Pohang University of Science and Technology, Korea) and an EG.7-OVA murine lymphoma cell was provided by Prof. Tae Heung Kang (Konkuk University, Korea). These cells were maintained in DMEM (GIBCO, Grand Island, NY, USA), supplemented with 10% fetal bovine serum (FBS) (GIBCO) and 1% penicillin/streptomycin (GIBCO), and incubated at 37 °C and 5% CO_2_. Cells were collected at 70–80% confluence and prepared for in vivo injection.

### 2.3. Generation and Activation of Mouse Bone-Marrow-Derived Dendritic Cells (BMDCs)

Bone marrow cells were isolated from femurs and tibiae of C57BL/6 mice, and red blood cells were removed using red blood cell lysis buffer (Sigma-Aldrich Inc., St. Louis, MO, USA). The bone marrow cells were differentiated into immature dendritic cells in DMEM medium supplemented with 10% FBS, 1% penicillin-streptomycin, and 50 μM 2-mercaptoethanol (M3148, Sigma-Aldrich) in the presence of GM-CSF (Peprotech, Cranbury, NJ, USA) (20 ng/mL) and IL-4 (Peprotech, Cranbury, NJ, USA) (10 ng/mL) for 7 days. The medium containing GM-CSF and IL-4 was exchanged after 4 days. To activate BMDCs with various adjuvants, BMDCs (2 × 10^6^ cells/mL) were stimulated with Pam3CSK4 (Invivogen, San Diego, CA, USA) (50 μg/mL), PolyI:C (Invivogen, San Diego, CA, USA) (40 μg/mL), or L-pampo™ at 37 °C for 24 h. L-pampo™ was treated with 1 dose, according to our standard operation procedure (SOP).

### 2.4. Adjuvant Formulation and Immunization

Female C57BL/6 mice were randomly assigned to each group and subcutaneously immunized with ovalbumin peptide (OVA_257–264_) or ovalbumin (OVA) formulated with various adjuvants. OVA_256–264_ peptide (SIINFEKEL) is an MHC class I (H-2Kb)-restricted peptide epitope of OVA and obtained from AnaSpec, Inc. (Fremont, CA, USA). OVA protein was obtained from Invivogen (San Diego, CA, USA). Montanide ISA 51 was obtained from SEPPIC (Paris, France). Murine recombinant GM-CSF was obtained from Peprotech (Cranbury, NJ, USA). Alum was purchased from (Invivogen, San Diego, CA, USA). All vaccine formulations were prepared in a dose of 100 μL per mouse. For peptide cancer vaccine, OVA_257–264_ peptide (50 µg/mouse) was dissolved in PBS and mixed with Pam3CSK4 (75 μg/mouse), Poly (I:C) (60 μg/mouse), murine recombinant GM-CSF (5 μg/mouse), and Montanide ISA 51 or L-pampo™ (1 dose). The Montanide ISA 51 was mixed with OVA_257–264_ peptide in 1:1 ratio (v/v). For the protein cancer vaccine, OVA (100 ug/mouse) was admixed with alum (100 µg) or L-pampo™ (1 dose). Peptide or protein formulated with adjuvants were administered to mice three times weekly. Splenocytes were harvested at day 7 after the last immunization and analyzed OVA peptide-specific T-cell responses.

### 2.5. Isolation of Lymph Node Cells from Inguinal Lymph Nodes

OVA_257–264_ peptide or OVA admixed with various adjuvants were subcutaneously injected on the right flank of C57BL/6 mice. After 24 h, the draining and non-draining inguinal lymph nodes (each one per mouse) were carefully removed, and their photographs were taken for the comparison of lymph node size. To isolate lymph node cells, lymph nodes were crushed through 70 µm strainers and resuspended in staining buffer or PBS. Lymph node cells from four lymph nodes within the group were pooled and counted for the following analyses.

### 2.6. Flow Cytometry

Single-cell suspensions were stained with the following antibodies: from Biolegend (San Diego, CA, USA), PerCP/Cyanine5.5-labeled anti-mouse CD11b (Clone M1/70), Brilliant Violet 421-labeled anti-mouse CD11c (Clone N418), PE/Cyanine7-labeled anti-mouse CD14 (Clone sa14-2), PE-labeled anti-mouse CD24 (Clone M1/69), Alexa Fluor 700-labeled anti-mouse CD45.2 (Clone 104), PE/Cyanine7-labeled anti-mouse CD86 (Clone GL-1), FITC-labeled anti-mouse H-2K^b^ (Clone AF6–88.5), APC-Cyanine7-labeled anti-mouse I-A/I-E (Clone M5/114.15.2), Brilliant Violet 650^TM^-labeled anti-mouse XCR1 (Clone ZET), and APC-labeled anti-mouse TNF-α (clone MP6-XT22) antibodies; from Invitrogen (Waltham, MA, USA), APC-labeled anti-mouse CD45RB220 (Clone RA3-6B2), APC-labeled anti-mouse CD3e (Clone 145-2C11), APC-labeled anti-mouse CD8a (Clone 53-6.7), FITC-labeled anti-mouse CD80 (Clone 16-10A1), andPE-labeled anti-mouse IFN-γ (Clone XMG1.2) antibodies; from ebioscience, PE-labeled anti-mouse CD8 (Clone 53-6.7) and FITC-labeled anti-mouse IFN-γ antibodies (Clone XMG1.2).

CD16/CD32 monoclonal antibody (93, eBioscience) was used to block the non-specific binding to the Fc receptor before surface staining. Cells were stained with LIVE/DEAD^TM^ Fixable Yellow Dead Cell Stain Kit (Invitrogen) for the detection of live/dead cells.

For the analysis of BMDC maturation, cells were stained with antibodies specific to CD40, CD80, and MHC class II at 4 °C for 30 min. To analyze DCs from lymph nodes, lymph node cells were stained with antibodies specific for CD11c, MHC class II, XCR1, CD11b, CD8a, B220, CD80, or CD86 at 4 °C for 30 min.

For intracellular cytokine staining, isolated splenocytes were stimulated with OVA_257–264_ peptide (1 µg/mL) at 37 °C for 16 h in the presence of 10 µg/mL brefeldin A (BD Biosciences, San Jose, CA, USA). Cells were subsequently stained with anti-CD8 antibody, fixed, and permeabilized with cytofix/cytoperm buffer and perm/wash buffer (BD), respectively, according to the manufacturer’s instruction (BD Biosciences, San Jose, CA, USA). Then, cells were stained with anti-IFN-γ and anti-TNF-α antibodies at room temperature for 30 min. Cells were stained in parallel with the respective control isotype antibodies.

Stained cells were acquired on a CytoFLEX (Beckman Coulter, Brea, CA, USA) or a BD Calibur (BD, Franklin Lakes, NJ, US) and analyzed with Flowjo software (v10.8.1) (Tree Star, Ashland, OR, USA).

### 2.7. T-Cell Proliferation Assay

Splenic CD3^+^ T cells were isolated using a Pan T cell isolation kit II (Miltenyi Biotec, Bergisch Gladbach, Germany) according to the manufacturer’s instructions. Briefly, splenocytes were incubated with CD3^+^ cell biotinylated antibody and anti-biotin microbeads kits. CD3^+^ T cells were isolated by using an LS MACS columns and the QuadroMACS system (Miltenyi Biotec). Sorted CD3^+^ T cells were incubated with CFSE (72782, Cell signaling technology, Danvers, MA, USA) at 37 °C for 15 min and washed with DMEM containing 10% FBS. CFSE-labeled CD3^+^ T cells (number) were co-cultured with BMDCs stimulated with various adjuvants for 72 h. The frequency of cell division of CD3^+^ T cells was determined using flow cytometry.

### 2.8. ELISA Assay

BMDCs (2 × 10^6^ cells/mL) were stimulated with Pam3CSK4 (50 μg/mL), PolyI:C (40 μg/mL), or L-pampo™ (1 dose) for 24 h. Non-stimulated BMDCs were used as a negative control. The culture supernatants were collected, and IL-12p70, TNF-α, and IL-6 productions were detected with ELISA kits according to the manufacturer’s instructions (BD Biosciences, San Jose, CA, USA).

### 2.9. Enzyme-Linked Immunospot (ELISpot) Assay

The OVA peptide-specific IFN-γ producing cell was assessed using the Mouse IFN-γ ELISpot kit according to the manufacturer’s protocol (3321-4HPW-10, Mabtech, Stockholm, Sweden). Briefly, immunized mouse splenocytes (5 × 10^5^ cells/100 μL) were seeded to a 96-well ELISpot plate pre-coated with anti-mouse IFN-γ antibodies and incubated with OVA_257–264_ peptide (25 μg/mL) at 37 °C for 20 h. After stimulation, the cells were removed, and the wells were incubated with biotinylated anti-mouse IFN-γ antibody at room temperature for 2 h. After the wells were washed, they were incubated with streptavidin-HRP for 1 h, and spot-forming cells (SFCs) were developed with TMB substrate solution. The spots were counted using an AELVIS ELISPOT Reader. To quantify peptide-specific responses, the mean spots of negative controls were subtracted from peptide-stimulated samples. The results were represented as SFCs/5 × 10^5^ cells.

### 2.10. Tumor Models and Treatment

B16F10-OVA murine melanoma cells (2 × 10^5^ cells/100 µL) or EG.7-OVA murine lymphoma cells (1 × 10^6^/100 µL) were subcutaneously inoculated into the right flank of C57BL/6 mice. Mice were immunized twice with OVA or OVA peptide admixed with L-pampo on days 3 and 10. For an immune checkpoint inhibitor treatment, anti-PD-L1 antibodies (BE0146, BioXcell, Lebanon NH, USA) (200 µg/dose) were administered via intraperitoneal injection at the indicated time points. Tumor size was measured with a caliper every three to four days, and survival was monitored. Tumor volumes were calculated using the equation (a^2^ × b)/2 (a, width; b, length). Mice are euthanized when exhibiting signs of poor health or when the tumor volume exceeded 2000 mm^3^.

### 2.11. Statistical Analysis

Data were analyzed using GraphPad Prism (Version 9) (San Diego, CA, USA). Data are represented as mean values ± standard error of the mean (SEM), as noted. To compare more than two groups, one-way ANOVA with Tukey’s test or Dunnett’s test was performed. Survival data were analyzed with Kaplan–Meier survival analysis. Differences of *p* < 0.05 were considered statistically significant.

## 3. Results

### 3.1. L-Pampo™ Promotes BMDC Maturation and T-Cell Proliferation

Dendritic cells (DCs) are professional antigen-presenting cells linking innate and adaptive immunity. After recognizing antigens, they mature by changing their phenotypes and functions and promoting antigen-specific adaptive immune responses [13]. To investigate whether L-pampo™ directly induces DC maturation and activation, we generated mouse bone-marrow-derived dendritic cells (BMDCs). We analyzed the expression levels of maturation markers on BMDCs upon stimulation with L-pampo™, TLR2 ligand (Pam3), or TLR3 ligand poly (I:C). L-pampo™ increased the expression of CD40, CD80, and MHC class II on BMDCs compared to Pam3 or poly (I:C) alone (Figure 1A–C). Activated DCs produce cytokines to induce the differentiation of T-cell populations. Then, we examined the cytokine production of DCs stimulated with L-pampo™, Pam3, or Poly (I:C). DCs stimulated with L-pampo™ significantly produced higher levels of IL-12p70, TNF-α, and IL-6 than Pam3- and poly (I:C)-only groups (Figure 1D–F). To determine whether L-pampo™-activated DCs elicit T-cell activation, we co-cultured DCs stimulated with L-pampo™ with CFSE-labeled splenic CD3^+^ T cells. DCs activated with L-pampo™ induced proliferative T-cell populations compared to control DCs and DCs stimulated with poly (I:C) alone (Figure 1G). These results demonstrate that L-pampo™ directly induces DC maturation and T-cell proliferation through activated DCs.

### 3.2. L-Pampo™ Promotes the Recruitment and Activation of DCs in Draining Lymph Nodes

Draining lymph nodes (dLNs), primary sites of antigen presentation, and antigen-specific T-cell priming are essential to maximize vaccine efficacy [14]. To examine the effect of L-pampo™ on the migration of DCs into the lymph nodes, we formulated OVA_257–264_ peptides with various adjuvants, including L-pampo™ and Montanide ISA 51 or GM-CSF, which are used in clinical trials of cancer peptide vaccines [15]. We then administered mice subcutaneously (s.c.) and analyzed DCs in the dLN using flow cytometry (Figure 2A and Appendix A). We observed that L-pampo™ adjuvanted formulation enlarged the size of dLNs compared to non-draining lymph nodes (ndLNs) (Appendix A). Given that the total CD45^+^ immune cells increased in dLNs but not ndLNs in all groups (Appendix A), we analyzed and compared lymph node cells in the dLNs of all groups. We found that OVA peptide admixed with L-pampo™ increased the mean number of total lymph node cells and the fold change to PBS compared with PBS, OVA peptide, Montanide ISA 51, and GM-CSF groups (Figure 2B).

We next analyzed the number of DCs and the levels of maturation markers on DCs in dLN. The L-pampo™ adjuvanted OVA peptide group was likely to increase the frequency and the number of DCs in dLN (Figure 2C and Appendix A). The fold change of DC in the L-pampo™ group significantly increased in dLN compared to that of the PBS, peptide-only, and GM-CSF groups but was not different from Montanide ISA 51 (Figure 2C). In addition, the OVA peptide admixed with the L-pampo™ group increased the levels of maturation markers, including CD80 and CD86, on DCs in dLN (Figure 2D,E). Then, we examined if L-pampo™ affects other myeloid cells that express TLR2 or TLR3, specifically the macrophage population (CD11c^−^CD11b^+^MHCII^+^), using a flow cytometric gating strategy (Appendix A). We found that OVA peptide admixed with the L-pampo™, Montanide ISA 51, and GM-CSF did not affect the macrophage population compared to PBS and peptide-only groups (Appendix A). These data suggest that L-pampo enhances DC migration and induces DC maturation within lymph nodes with less impact on the macrophage population.

We also evaluated the effect of L-pampo™ for protein vaccines on the DC recruitment and maturation in dLN. We formulated OVA protein (OVA) with L-pampo™ or alum, the most widely used adjuvant for protein subunit vaccine, and analyzed DC migration and activation in the dLN using flow cytometry. OVA formulated with L-pampo™ increased the mean number of total CD45^+^ immune cells and DCs in the dLN compared to the control and alum groups (Appendix A). Moreover, OVA admixed with L-pampo™ enhanced the expression of CD80 on DC in dLN compared to the alum group (Appendix A). These data demonstrate that peptide- or protein-admixed L-pampo™ directly promotes DC migration into dLN and induces DC maturation that facilitates T-cell priming and activation within the lymph nodes.

### 3.3. L-Pampo™ Increases DC Subsets in Draining Lymph Nodes

DCs are classified as conventional DCs (cDCs) and plasmacytoid DCs (pDCs) based on their phenotype and function [16]. cDCs can be further subdivided into cDC1 (XCR1^+^) and cDC2 (CD11b^+^), which induce Th1 and Th2 immune responses, respectively. pDCs (B220^+^) are specialized in producing type I IFN that activates the CD8^+^ T-cell anti-tumor response [17]. For effective T-cell-mediated anti-tumor immunity, DCs need to cross-present tumor antigen to CD8^+^ T cells [18]. Among DC subsets, CD8a^+^ cDC1s are the most efficient in antigen cross-presentation and the induction of efficient cytotoxic T-cell immune response [19]. Therefore, we sought to determine if L-pampo™ affects the distribution of DC subsets in dLNs. We immunized mice with OVA_257–264_ peptide mixed with various adjuvants and analyzed DC subsets in dLNs using flow cytometry (Figure 2A and Appendix A). We observed that L-pampo™ increased the frequency and the fold change of cDC1s and cross-presenting DCs compared to the PBS, peptide-only, Montanide ISA 51, and GM-CSF groups (Figure 2F,G and Appendix A). cDC2s and pDCs also increased in the L-pampo™ groups than in PBS, peptide-only, and GM-CSF groups (Figure 2H,I and Appendix A). Consistent with the results of OVA peptide formulation, OVA admixed with L-pampo™ augmented cDC1s, cross-presenting DCs, cDC2s, and pDCs in dLNs compared to the PBS, OVA-only, and OVA-alum formulation groups (Appendix A). These findings indicate that L-pampo™ adjuvanted formulations expand DC subsets in dLNs that are critical for anti-tumor T-cell immunity.

### 3.4. L-Pampo™ Increases OVA Peptide-Specific T-Cell Responses and Inhibits Tumor Growth

To determine if the L-pampo™ adjuvanted peptide vaccine elicits antigen-specific T-cell responses in a mouse model, we immunized mice with OVA_257–264_ peptides admixed with L-pampo™, Montanide ISA 51, or GM-CSF. We analyzed antigen-specific T-cell responses using IFN-γ ELISPOT assay (Figure 3A). We found that the L-pampo™ group significantly increases the number of IFN-γ-producing cells compared to PBS and OVA peptide groups, while Montanide ISA51 and GM-CSF groups did not induce OVA-specific T-cell responses (Figure 3B).

We then investigated the anti-tumor efficacy of OVA peptide combined with L-pampo™ in a B16F10 mouse melanoma model. We subcutaneously inoculated B16F10-OVA cells in mice and administered OVA-peptide with L-pampo™, Montanide ISA 51, or GM-CSF on days 3, 10, and 17 (Figure 4A). We observed that the L-pampo™ group significantly reduced tumor growth compared to the PBS group. In contrast, Montanide ISA 51 and GM-CSF groups were likely to reduce tumor growth but not significantly compared to control groups (Figure 4B). Similarly, the L-pampo™ group improved survival rates compared to PBS and OVA-peptide groups (Figure 4C), indicating that L-pampo™ can be an adjuvant for a peptide cancer vaccine to induce anti-tumor efficacy.

### 3.5. Combination of OVA Admixed with L-Pampo™ with Immune Checkpoint Inhibitors Enhances Antigen-Specific T-Cell Response and Anti-Tumor Efficacy

Immune checkpoint inhibitors (ICIs), such as CTLA-4, PD-1, and PD-L1 inhibitors, are a revolutionary cancer immunotherapy that improves treating a wide range of cancers [20]. However, ICIs still have limitations such as primary and acquired resistance, severe toxicity, and non-responders to ICIs [21]. To overcome the limitation and improve clinical outcomes, combining cancer vaccine and ICI therapy has emerged as a promising therapeutic approach to maximize the anti-tumor effect [22]. To determine whether a combination therapy of L-pampo™ adjuvanted vaccine with ICI increases the antigen-specific T-cell immune responses, we formulated OVA with L-pampo™ and immunized mice on days 0 and 7 followed by the treatment of PD-L1 inhibitor via intraperitoneal injection on days 8, 10, and 12 (Figure 5A). We detected OVA-specific T-cell immune responses using flow cytometric analysis. As expected, the L-pampo™ admixed OVA group increased the number of IFN-γ^+^ CD8^+^ T cells, TNF-α^+^ CD8^+^ T cells, and IFN-γ^+^ TNF-α+ CD8^+^ T cells compared to PBS, OVA-only, and PD-L1 antibody (Ab) groups (Figure 5B and Appendix A). Interestingly, the combination of L-pampo™ with the PD-L1 Ab group synergistically increased antigen-specific T-cell responses compared to PBS, OVA-only, PD-L1 Ab, and L-pampo™ groups (Figure 5B), suggesting that L-pampo™ cancer vaccine and its combination with ICIs can bolster CD8^+^ T-cell responses that are critical for tumor treatment.

Next, we evaluated the anti-tumor efficacy of L-pampo™ or L-pampo™ and ICI combination therapy in an EG.7 mouse tumor model. We subcutaneously injected EG.7-OVA tumor cells into the mice and then immunized OVA admixed with L-pampo™ on days 3 and 10. We injected PD-L1 inhibitors five times every other day starting on day 11 and accessed the tumor volume and survival rate (Figure 6A). We observed that the PD-L1 antibody-treated group reduced the tumor volume compared to PBS and OVA-only groups. The L-pampo™ group significantly regressed tumor growth compared to the PBS, OVA-only, and PD-L1 Ab groups. The combination of the L-pampo™ adjuvanted vaccine with the PD-L1 inhibitor nearly eliminated tumor growth (Figure 6B). Consistent with the data about the tumor volume, the combination of L-pampo™ with the PD-L1 inhibitor group improved the survival rate compared to the control and PD-L1 groups (Figure 6C), supporting the notion that L-pampo™ can be a potent cancer vaccine adjuvant and a synergistic partner of ICIs for cancer treatment.

## 4. Discussion

In this study, we demonstrated the efficacy of L-pampo™ as a potent cancer vaccine adjuvant that directly activates DCs and triggers their migration to draining lymph nodes where antigen presentation and T-cell priming occur. Moreover, a cancer vaccine formulated with L-pampo™ promoted antigen-specific T-cell immune responses and anti-cancer efficacy in murine tumor models. Notably, combining the L-pampo™ adjuvanted cancer vaccine with immune checkpoint inhibitors (ICIs) enhanced tumor regression and long-term survival benefits. Thus, L-pampo™ can be a promising cancer vaccine adjuvant and a potent candidate for combination cancer therapy.

In general, adjuvants can be classified into two types based on their functions: immune-stimulating adjuvants and depot-forming adjuvants. Montanide ISA 51, which is a representative depot-forming adjuvant, enhances innate and adaptive immunity by protecting and slowly releasing the antigen over time. This feature benefits peptide-based cancer vaccines with limitations such as peptide degradation and short half-life, and many clinical trials have used Montanide ISA 51 as a peptide cancer vaccine adjuvant [5]. However, recent studies have reported that Montanide ISA 51 often causes local side effects such as skin irritation, inflammation, and even ulcers and that Montanide ISA 51 can lead to local sequestration that impairs adaptive immunity in lymph nodes and causes effector T-cell dysfunction [23,24]. Our study demonstrated that Montanide ISA 51 failed to induce sufficient DC migration, DC maturation, and expansion of cDC1 and cross-presenting DCs. This result may be attributed to Montanide ISA 51′s depot effect that hinders the efficient DC migration into draining lymph nodes, leading to a reduction in anti-tumor immunity and efficacy.

GM-CSF is a cytokine that elicits anti-tumor immune responses in cancer vaccine formulation. Several studies have reported that vaccines formulated with GM-CSF increased anti-cancer efficacy in patients with NSCLC and pancreatic cancer [25,26]. However, our study’s peptide vaccine formulated with GM-CSF promoted neither DC recruitment nor DC activation. Consequently, this vaccine formulation failed to induce sufficient T-cell responses and anti-tumor efficacy. Growing evidence supports the claim that the anti-tumor efficacy of GM-CSF was contradictory in several clinical trials [6]. Peptide-GM-CSF clinical trials demonstrate that GM-CSF affects local and systemic toxicity but not immune responses [27]. Moreover, a large body of experimental evidence suggests that GM-CSF has a pro-tumorigenic role. GM-CSF can promote cancer progression by inducing pro-tumorigenic immune cells such as myeloid-derived suppressor cells (MDSCs), tumor-associated macrophages (TAMs), and Treg cells [6,28]. GM-CSF can also convert DCs to tolerogenic phenotypes that impair antigen presentation and T-cell functions [29]. A recent study suggested that a modified immunization regimen, consisting of GM-CSF pretreatment for three days and vaccination, induced sufficient DC expansion and robust antigen-specific immune response [30]. Hence, further investigation is needed to determine whether this modified immunization regimen improves GM-CSF’s anti-tumor function and whether the combination of GM-CSF and L-pampo™ promotes robust immune response and anti-cancer efficacy.

Immune checkpoint inhibitors (ICIs) have achieved improved clinical benefits. However, a large subset of patients does not benefit from ICI therapy. Most cancers have a 15–30% objective response rate, and “immune-cold” solid tumors such as pancreatic, glioma, and colon cancer are utterly non-responsive to ICIs [31]. The low response rate observed in most cancers may be attributed to (1) insufficient generation of anti-tumor T cells, (2) dysfunction of tumor-specific T cells [32], and (3) lack of T-cell memory [33,34]. Thus, a combination of a cancer vaccine, which can elicit specific T-cell responses, and ICIs can be an attractive therapeutic option. Our study showed that L-pampo™ alone induced sufficient T-cell responses and improved anti-cancer efficacy. More interestingly, combining a cancer vaccine with L-pampo™ and anti-PD-L1 antibody bolstered robust CD8^+^ T-cell responses and enhanced anti-cancer efficacy, suggesting the possibility of overcoming low response rates and improving patients’ outcomes with cold tumor therapy. Further studies will explore if the L-pampo™-containing vaccine can also synergize with other ICIs, such as an anti-CTLA-4 antibody, if this combination works in other tumor models such as prostate or colon cancer, and if L-pampo™, alone or in combination with ICI, affects T-cell memory.

## 5. Conclusions

Our study suggests that L-pampo™, a novel TLR agonist, can be a promising cancer vaccine adjuvant that modulates DC migration and activation in draining lymph nodes. Our findings also provide implications for the merit of combining cancer vaccines with ICIs to overcome the non-responsiveness to ICI and ultimately achieve clinical benefits for patients with immune-cold tumors.

## Figures and Tables

**Figure 1 cancers-15-03978-f001:**
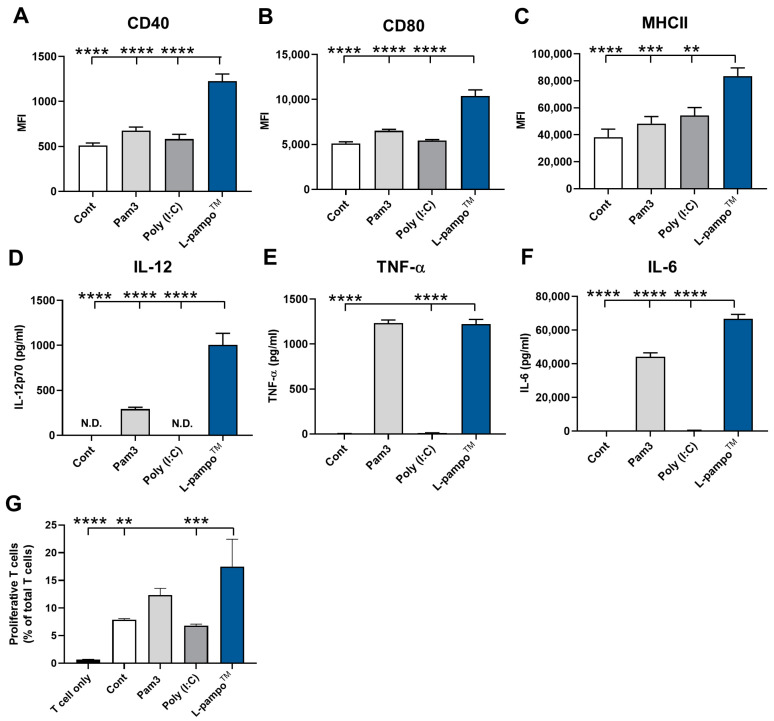
L-pampo™ promotes BMDC maturation and T-cell proliferation. (**A**–**F**) BMDCs were stimulated with Pam3, Poly (I:C), or L-pampo™ for 24 h. (**A**) CD40, (**B**) CD80, and (**C**) MHC class II on BMDCs were analyzed through flow cytometry. (**D**) IL-12p70 (**E**) TNF-α, (**F**) IL-6 production in the supernatants of BMDC were analyzed using ELISA. (**G**) T-cell proliferation assay. BMDCs were stimulated with Pam3, Poly (I:C) or L-pampo™ for 24 h. CFSE-labeled splenic CD3^+^ T cells from syngeneic mice were coincubated for 3 days with BMDCs stimulated with adjuvants. Data (bars) indicate mean ± SEM. Data are representative of three independent experiments. ** *p* < 0.01, *** *p* < 0.001, **** *p* < 0.0001, one-way ANOVA with Tukey’s test.

**Figure 2 cancers-15-03978-f002:**
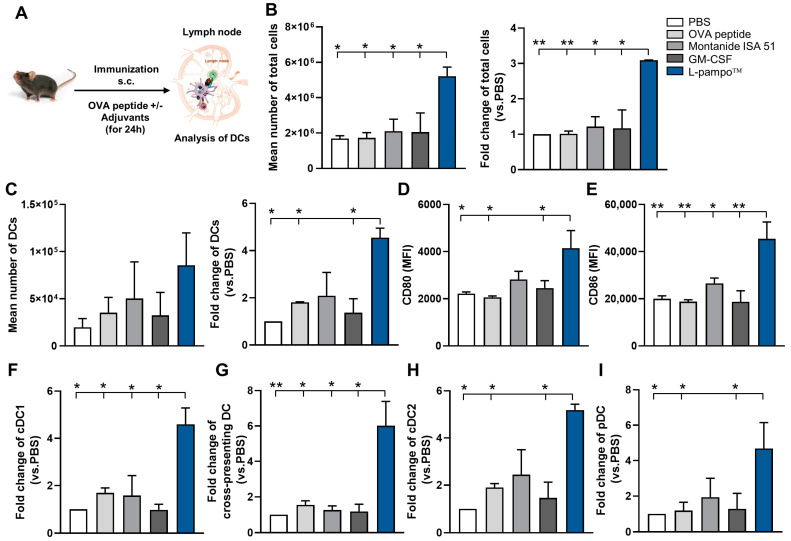
L-pampo™ promotes DC recruitment and maturation and increases DC subsets in dLNs. C57BL/6 mice (total *n* = 40; *n* = 8/group) were s.c. injected with OVA_257–264_ peptide and/or various adjuvants, including L-pampo™, Montanide ISA 51, and GM-CSF, and then inguinal lymph nodes were harvested at 24 h after injection. (**A**) A schematic detailing of the in vivo experiment. (**B**) The mean total CD45^+^ immune cell numbers from the four draining lymph nodes (**left**) and the fold changes to PBS (**right**). (**C**) The numbers of DC (CD11c^+^ MHCII^high^) from the four draining lymph nodes (**left**) and the fold changes to PBS (**right**). (**D**) MFI (mean fluorescence of intensity) of CD80 on DC population. (**E**) MFI of CD86 on DCs. (**F**–**I**) DC subsets in lymph nodes were analyzed using flow cytometry and shown as the fold change to PBS. (**F**) cDC1s (CD11c^+^ MHCII^high^ XCR1^+^). (**G**) cross-presenting DCs (CD11c^+^ MHCII^high^ CD8a^+^). (**H**) cDC2s (CD11c^+^ MHCII^high^ CD11b^+^). (**I**) pDCs (CD11c^+^ MHCII^high^ CD45B220^+^). Data (bars) represent mean ± SEM. Data reflect two independent experiments. Asterisks indicate statistically significant differences in comparison to the L-pampo™ group. * *p* < 0.05, ** *p* < 0.01, one-way ANOVA with Dunnett’s test.

**Figure 3 cancers-15-03978-f003:**
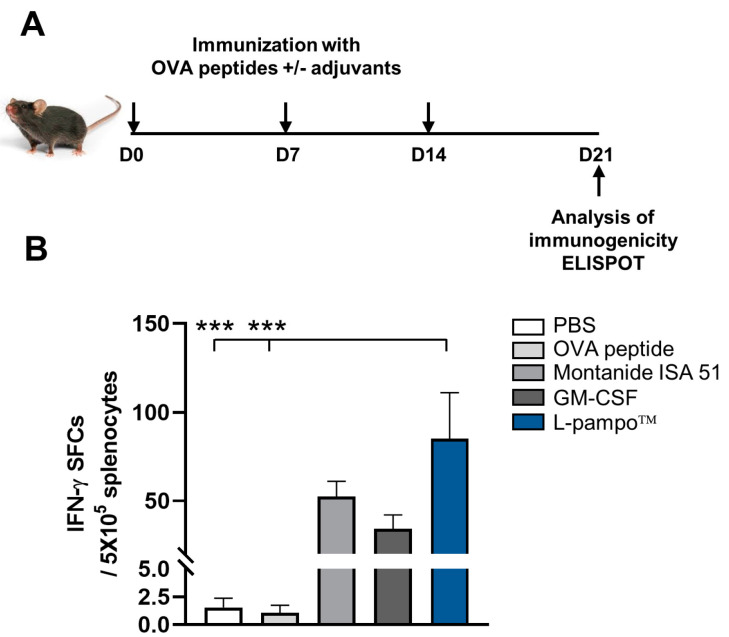
L-pampo™ increases OVA peptide-specific T-cell responses. C57BL/6 mice (total *n* = 71; *n* = 13/PBS or OVA peptide; *n* = 15/Montanide ISA 51, GM-CSF, or L-pampo™) were s.c.-injected with OVA_257–264_ peptide and/or L-pampo™, Montanide ISA 51, or GM-CSF on days 0, 7, and 14. Spleens were harvested on day 21, and T-cell responses were analyzed using ELISPOT assay. (**A**) A schematic of immunization strategy. (**B**) The number of OVA_256–264_ peptide-specific IFN-γ secreting cells. Data (bars) represent mean ± SEM. Data reflect three independent experiments. *** *p* < 0.001, one-way ANOVA with Tukey’s test.

**Figure 4 cancers-15-03978-f004:**
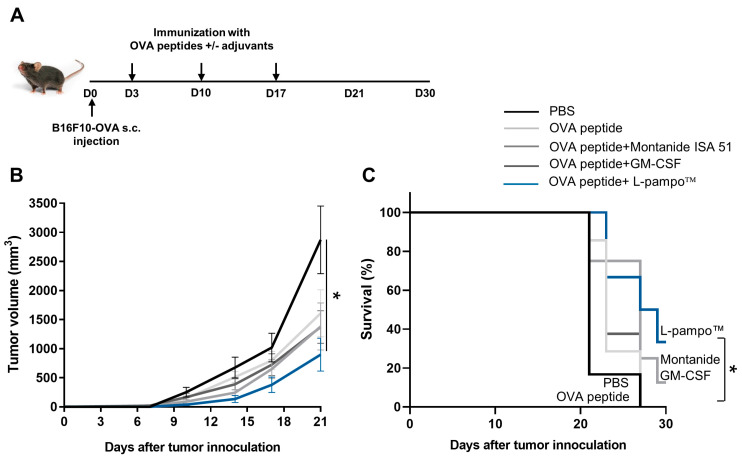
OVA peptide admixed with L-pampo™ increases anti-tumor efficacy. C57BL/6 mice (total *n* = 36; *n* = 6/PBS; *n* = 7/OVA peptide or L-pampo™; *n* = 8/Montanide ISA 51 or GM-CSF) were subcutaneously inoculated with B16F10-OVA tumor cells (2 × 10 ^5^/100 µL) and then subcutaneously immunized with OVA_257–264_ peptide admixed with L-pampo™, Montanide ISA 51 or GM-CSF on days 3, 10, and 17. Tumor volumes were measured every three or four days using calipers, and survival was monitored. (**A**) A schematic of tumor inoculation and immunization strategy. (**B**) Tumor volumes. (**C**) Survival rate (%) of tumor-bearing mice. Data represent mean ± SEM. Data reflect two independent experiments. * *p* < 0.05, one-way ANOVA with Tukey’s test (**B**), Kaplan–Meier survival analysis (**C**).

**Figure 5 cancers-15-03978-f005:**
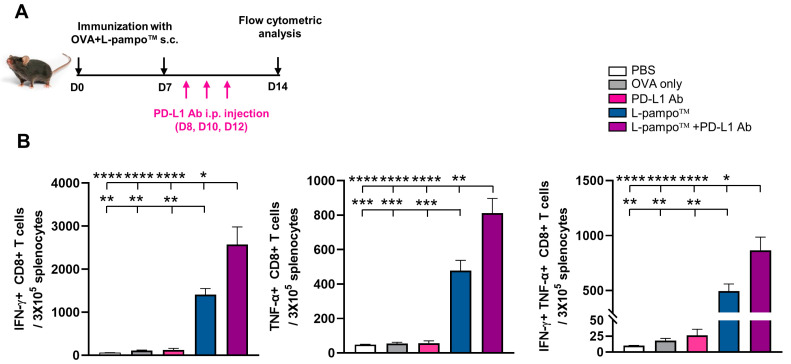
OVA admixed with L-pampo™ or combinations with PD-L1 inhibitor induces antigen-specific T-cell immune responses. C57BL/6 mice (total *n* = 15; *n* = 3/group) were subcutaneously immunized with OVA-admixed with L-pampo™ on days 0 and 7. PD-L1 antibodies were administered to immunized mice using intraperitoneal injection (i.p) on days 8, 10, and 12. Spleens were harvested on day 14, and cytokine-producing CD8^+^ T cells were detected using flow cytometry analyses. (**A**) A schematic of immunization and ICI administration strategy. (**B**) The number of OVA_256–264_ peptide-specific IFN-γ, TNF-α or IFN-γ and TNF-α producing CD8^+^ T cells. Data (bars) represent mean ± SEM. Data are representative of two independent experiments. * *p* < 0.05, ** *p* < 0.01, *** *p* < 0.001, **** *p* < 0.0001, one-way ANOVA with Tukey’s test.

**Figure 6 cancers-15-03978-f006:**
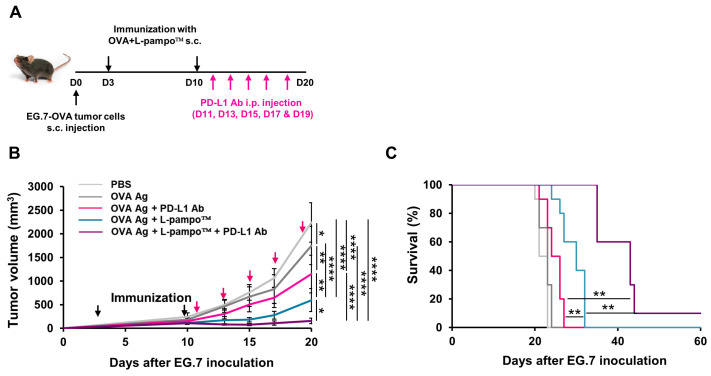
OVA admixed with L-pampo™ or in combination with PD-L1 inhibitor enhances anti-cancer efficacy. C57BL/6 mice (total *n* = 50; *n* = 10/group) were subcutaneously inoculated with EG.7-OVA tumor cells (1 × 10^6^/100 µL) and then immunized with OVA-admixed with L-pampo™ on days 3 and 10. PD-L1 antibodies were administered to immunized mice using intraperitoneal injection (i.p) five times every two days. Tumor volumes were measured every two or three days, and survival was monitored. (**A**) A schematic of tumor injection, immunization, and ICI administration strategy. (**B**) Tumor volumes were measured at indicated time points. (**C**) Survival rate (%) of tumor-bearing mice. Dark and magenta arrows represent the time points of immunization and PD-L1 antibody injection, respectively. Data reflect two independent experiments. Data are shown as mean ± SEM (**B**). * *p* < 0.05, ** *p* < 0.01, **** *p* < 0.0001, one-way ANOVA with Tukey’s test (**B**), Kaplan–Meier survival analysis (**C**).

## Data Availability

The data can be shared up on request.

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
