# Peer review of "L-Pampo™, a Novel TLR2/3 Agonist, Acts as a Potent Cancer Vaccine Adjuvant by Activating Draining Lymph Node Dendritic Cells"

_cancers, 2023, doi:10.3390/cancers15153978_

Round 1

Reviewer 1 Report

In this manuscript, Yoonki Heo et.al. investigated that L-pampo™ modulated DC migration and activating in draining lymph nodes to promote antigen-specific T-cell responses and anti-tumor efficacy. Moreover, combining vaccines formulated with L-pampo™ with immune checkpoint inhibitors induce a synergistic anti-tumor effect, suggesting that L-pampo™ can be a potent cancer vaccine adjuvant. It's known that the TLR2 agonist Pam3CSK4 binds to the TLR3 agonist Poly (I:C) to generate L-pampo™. For the manuscript, the stimulated activation of dendritic cells and CD8+ T cells by L-pampo™ as an adjuvant for tumor immunovaccines was well grounded and statistical methods were used appropriately. However, there are serious problems interpreting with experimental results and experimental design, so that the expression of the manuscript is not consistent with the figures.

1) The results of flow analysis only contain bar graphs of the analysis and lack       representative FACS plots.

2) In Figure 1G, regarding the T cell proliferation assay, is it necessary to additionally   isolate T cells in the draining lymph nodes and co-culture with BMDCs stimulated with various adjuvants to be more consistent with the later results of the recruitment of DCs in the draining lymph nodes related to T cell activation?

3) In Figure 2, the labeling of the subgroups should all be consistent with the labeling of the subgroups in Figure6B to make it clearer. OVA peptide should be supplemented before combining various adjuvants. In addition, the distribution of DC subsets in dLNs are mentioned here, but the functions of cytokines secreted by different DC subsets are not mentioned here.

4) In Figure 3, only the level of IFN-γ was tested to illustrate that L-pampo™ increases OVA peptide-specific T cell responses. Other indicators such as IFN-α and TNF-α, which indicate the activity and tumor-killing ability of CD8+ T cells, should be added.

5) Figure 3B does not show a statistical difference between the L-pampo™ group with the two groups, Montanide ISA 51 and GM-CSF groups, which is inconsistent with the conclusions in the text (line 352).

6) In Figure 4, only the change in tumor size after group treatment is shown, which is not consistent with the anti-tumor immunity in the caption of the figure4 note. In addition to tumor size, relevant experiments such as Survival rate of tumor-bearing mice and immune infiltration of tumors should be added.

7) The in vivo experiments throughout this manuscript fail to show that L-pampo™ increases T cell responses by inducing maturation and activation of DC cells and DC recruitment in the draining lymph nodes. Because it does not involve other immune cells besides DC cells that can be similarly activated by L-pampo™, it ignores the complexity of the tumor immune microenvironment and is insufficient to draw final conclusions.

Reviewer 2 Report

To Author:

In this article, the authors demonstrated that the L-pampo™ can activate DCs and promote T cell proliferation. L-pampo™ was also shown to inhibit tumor growth in this study. I considered this research article to be significant. However, I have several suggestions before it can be accepted.

 Comments:

(1) The authors should detect which downstream signaling pathway of TLR2/3 is activated by L-pampo™.

(2) In addition to the tumor volume (Figure 4B) when detecting the effect of L-pampo™ on tumor, the authors should also detect the changes in the histopathology of the major organs (liver, lung and brain).

(3) Tumor metastasis is a major cause of death of cancer patients, the authors should also detect whether L-pampo™ affects tumor metastasis.

(4) TLR2/3 is expressed in a variety of immune cells, and macrophages are also common infiltrating cells in tumor tissue. The authors should detect the effect of L-pampo™ on macrophages.

(5) The authors demonstrated that L-pampo™ can promote the proliferation of T cells (Figure 1G), but the authors did not detect the changes of immune cells (DCs, CD8+ T cell and macrophages) in tumor tissue when studying the effect of L-pampo™ on tumor.

Reviewer 3 Report

The present manuscript describes “L-pampoTM, a novel TLR2/3 agonist, acts as a potent cancer vaccine adjuvant by activating draining lymph node dendritic cells”. In the present study the authors demonstrated that vaccines formulated with L-pampoTM affect dendritic cells (DCs) recruitment and activation in draining lymph nodes (dLNs) and lead to antigen-specific T-cell responses and anti-tumor efficacy. To study L-PampoTM, the authors bought the female animals, murine melanoma cancer cell lines. To investigate L-pampoTM directly induces DC maturation and activation, the authors generated mouse bone marrow-derived dendritic cells (BMDCs). They analyzed the expression levels of maturation markers on BMDCs upon stimulation with L-pampoTM, TLR2 ligand (Pam3), or TLR3 ligand poly (I:C). L-pampoTM increased the expression of CD40, CD80, and MHC class II on BMDCs compared to Pam3 or poly (I:C) alone. The activated dendritic cells produce cytokines to induce differentiation of T cell populations. The authors also examined the cytokine production of DCs stimulated with L-pampoTM, Pam3, or Poly (I:C). DCs stimulated with L-pampoTM significantly produced higher levels of IL-12p70, TNF-α, and IL-6 than Pam3 and poly (I:C) only groups (Figure 1D-F). To determine whether L-pampoTM-activated DCs elicit T cell activation, the co-cultured DCs stimulated with L-pampoTM with CFSE-labeled splenic CD3T cells. DCs activated with L-pampoTM induced proliferative T cell populations compared to control DCs and DCs stimulated with poly (I:C) alone. These results demonstrate that L-pampoTM directly induces DC maturation and T-cell proliferation through activated DCs. The present study demonstrated that L-pampoTM, a novel TLR agonist, can be a promising cancer vaccine adjuvant by modulating dendritic cells migration and activating in draining lymph nodes. I would recommend to publish this manuscript in Cancers Journal. 

Author Response

Reviewer:

The present manuscript describes “L-pampoTM, a novel TLR2/3 agonist, acts as a potent cancer vaccine adjuvant by activating draining lymph node dendritic cells”. In the present study the authors demonstrated that vaccines formulated with L-pampoTM affect dendritic cells (DCs) recruitment and activation in draining lymph nodes (dLNs) and lead to antigen-specific T-cell responses and anti-tumor efficacy. To study L-PampoTM, the authors bought the female animals, murine melanoma cancer cell lines. To investigate L-pampoTM directly induces DC maturation and activation, the authors generated mouse bone marrow-derived dendritic cells (BMDCs). They analyzed the expression levels of maturation markers on BMDCs upon stimulation with L-pampoTM, TLR2 ligand (Pam3), or TLR3 ligand poly (I:C). L-pampoTM increased the expression of CD40, CD80, and MHC class II on BMDCs compared to Pam3 or poly (I:C) alone. The activated dendritic cells produce cytokines to induce differentiation of T cell populations. The authors also examined the cytokine production of DCs stimulated with L-pampoTM, Pam3, or Poly (I:C). DCs stimulated with L-pampoTM significantly produced higher levels of IL-12p70, TNF-α, and IL-6 than Pam3 and poly (I:C) only groups (Figure 1D-F). To determine whether L-pampoTM-activated DCs elicit T cell activation, the co-cultured DCs stimulated with L-pampoTM with CFSE-labeled splenic CD3+ T cells. DCs activated with L-pampoTM induced proliferative T cell populations compared to control DCs and DCs stimulated with poly (I:C) alone. These results demonstrate that L-pampoTM directly induces DC maturation and T-cell proliferation through activated DCs. The present study demonstrated that L-pampoTM, a novel TLR agonist, can be a promising cancer vaccine adjuvant by modulating dendritic cells migration and activating in draining lymph nodes. I would recommend to publish this manuscript in Cancers Journal.

Response: We sincerely appreciate the reviewer’s recommendation to publish our manuscript in the Cancers Journal.

Round 2

Reviewer 1 Report

I have no further questions on the revised manuscript.

Reviewer 2 Report

The revised paper is ok. I don't have any comments.